# Using a standardized sound set to help characterize misophonia: The International Affective Digitized Sounds

**Jacqueline Trumbull** [1], **Noah Lanier**[1], **Katherine McMahon**[2], **Rachel Guetta**[1], **M. Zachary Rosenthal**[1,2] *

**1** Duke University, Durham, NC, United States of America, **2** Duke University Medical Center, Durham, NC, United States of America

* mark.rosenthal@duke.edu

**Data Availability Statement:** The data files underlying the results presented in the study will be made available via the Duke Data Research Repository (https://research.repository.duke.edu);

## Abstract

Misophonia is a condition characterized by negative affect, intolerance, and functional impairment in response to particular repetitive sounds usually made by others (e.g., chewing, sniffing, pen tapping) and associated stimuli. To date, researchers have largely studied misophonia using self-report measures. As the field is quickly expanding, assessment approaches need to advance to include more objective measures capable of differentiating those with and without misophonia. Although several studies have used sounds as experimental stimuli, few have used standardized stimuli sets with demonstrated reliability or validity. To conduct rigorous research in an effort to better understand misophonia, it is important to have an easily accessible, standardized set of acoustic stimuli for use across studies. Accordingly, in the present study, the International Affective Digitized Sounds (IADS-2), developed by Bradley and Lang (Bradley MM et al., 2007), were used to determine whether participants with misophonia responded to certain standardized sounds differently than a control group. Participants were 377 adults (132 participants with misophonia and 245 controls) recruited from an online platform to complete several questionnaires and respond to four probes (arousal, valence, similarity to personally-relevant aversive sounds, and sound avoidance) in response to normed pleasant, unpleasant, and neutral IADS-2 sounds. Findings indicated that compared to controls, participants with high misophonia symptoms rated pleasant and neutral sounds as significantly more (a) arousing and similar to trigger sounds in their everyday life, (b) unpleasant and (c) likely to be avoided in everyday life. For future scientific and clinical innovation, we include a ranked list of IADS-2 stimuli differentiating responses in those with and without misophonia, which we call the IADS-M.

## Introduction

Misophonia is a recently defined disorder characterized by intolerance to specific sounds and associated stimuli [1]. Aversive sounds (also sometimes called triggers) are most typically repetitive and produced by others orally or facially (e.g., chewing, throat-clearing, sniffing),

data uploaded in the Duke Research Data Repository and a link to it here: https://g-4ab488. cfd00a.5898.data.globus.org/IADS_Data_Clean. sav.

**Funding:** The authors received no specific funding for this work.

though environmental noises also are common (e.g., pens clicking, clocks ticking) [2]. Misophonia is characterized by unusually strong multi-modal emotional responses to trigger sounds irrespective of the acoustic features of these stimuli (e.g., loudness or frequency), as well as acknowledgement that this reaction is disproportionate [3]. Though arguably under-recognized, it has recently gained attention in the popular press, and academic research has rapidly accelerated since Schröder, Vulink & Denys [4] in 2013 proposed that misophonia be considered a new psychiatric disorder and recommended diagnostic criteria.

Despite rapidly advancing research in the first 10 years of scientific studies examining misophonia, much remains scientifically unknown about the etiology, course, or underlying nature and features that differentiate it from other clinical presentations. Most of the early-stage research investigating misophonia has used questionnaires and other subjective measurement approaches. Responses to sounds presented in a standardized manner may help to assess misophonia more objectively and precisely than relying on self-report questionnaires alone. Once sound banks for misophonia are developed and validated, researchers could use them in conjunction with objective measures of biological, psychological, or social processes (e.g., psychophysiological measures, neuroimaging, etc.) to objectively characterize misophonic responses to trigger sounds in laboratory or treatment studies. Clinicians could use such sound banks to assist with clinical assessment and in certain interventions (e.g., inhibitory learning-based approaches) [5]. Further, using a more standardized measurement approach would help to more comprehensively characterize the nature and scope of trigger sounds and any shared underlying acoustic features.

In contrast, if a more standardized approach is not developed, research studies may rely only on asking participants to identify sounds most bothersome to them. This could result in the research literature remaining somewhat limited in discoveries about misophonia. More specifically, in the absence of standardized measurement using more objective methods, the knowledge base around misophonia may be constrained only to sounds that are most top of mind, or that participants report because the sounds are already widely known to be common in misophonia (e.g., chewing, eating), rather than sounds that are highly aversive but less commonly encountered or reported.

A number of studies have used recorded sounds to obtain a more objective measure of how individuals with misophonia respond to sounds in laboratory conditions. However, these studies largely have used unstandardized stimuli, with sound content chosen based on known triggers from the emerging research literature, and recordings taken from different sources, including YouTube and those created by researchers [6–13]. One recent study used a data driven approach to create a sound bank composed of standardized sounds [14], though sounds were chosen a priori from multiple sources with unknown psychometric properties.

Although these studies used acoustic stimuli, their methodologies have notable limitations. For example, selection of experimental stimuli in most studies has been limited to study team expertise and a review of the literature. Accordingly, most sound sets used were limited by a priori assumptions about the best sounds to use as triggers. On the one hand, it is appropriate to consider existing research and clinical experience in considering which sounds to use experimentally. On the other, if researchers only use their clinical experiences and the currently available research, the full scope of sounds that differentiate those with and without misophonia may be limited.

An additional limitation to most previous studies examining responsivity to sounds is that comparison sounds were not created using the same method as experimental (i.e., misophonia trigger) sounds, precluding clear interpretations of study findings. These studies typically selected sounds from multiple sources; the sounds themselves are of various lengths, are not controlled for sound qualities such as volume or duration and are largely not drawn from

sound sets with standardized norms, reliability, or validity. Ultimately, these sound sets tend to have high face validity but are unstandardized and have undetermined psychometric properties, collectively rendering study interpretations somewhat inconclusive.

Additionally, previous studies using recorded sounds have not used standardized and psychometrically validated sounds derived from basic emotion research. The Free Open-Access Misophonia Stimulus (FOAMS) database advances the literature by providing standardized sounds that are highly accessible to researchers [15]. However, FOAMS differs from the current study in how sounds were chosen and rated. Sounds in the FOAMS dataset were initially chosen a priori and do not come from a sound set that is commonly used in emotion research. Ratings were derived by asking participants to identify the sound and rate how bothersome it was to them, rather than more standardized ratings such as valence and arousal. Sounds standardized on emotional arousal and valence may be particularly useful for contextualizing misophonia in comparison to the literature on other auditory or psychiatric conditions involving sound sensitivities that have used valence and arousal ratings in studies. Using these standardized ratings can also provide greater understanding of misophonia, as the disorder is widely characterized as featuring physiological (e.g., sympathetic nervous system activation), behavioral (e.g., avoidance, escape, and verbal aggression), and subjective (e.g., certain internal or external attributions; affective states such as anger, anxiety, or moral disgust) emotional responses when anticipating or reacting to personally-relevant aversive cues [1].

An additional consideration germane to misophonia is the degree to which one perceives an experimental sound to be similar to their own personally relevant triggers. To this purpose, researchers could use sound stimuli that are created in a non-standardized way, tailored to each participant. An advantage of that approach is that the sounds are personally relevant to each participant. However, sound responses across patients may be confounded by individual differences in the acoustic properties of the sounds (e.g., volume, duration). In effect, that approach would sacrifice internal validity for limited external validity. One possible solution to this challenge is to identify standardized sounds that are also experienced as personally relevant. To do this, participants can rate sounds on the similarity to their own trigger sounds.

A final consideration when identifying standardized sounds to use in misophonia is the degree to which these sounds elicit behavioral responses consistent with what would be expected in those with the condition. Avoidance and escape behaviors are widely characterized as primary responses to anticipated contexts or triggers [4], yet no studies have included ratings of how likely a participant would be to avoid a particular trigger sound.

Creating a misophonia-specific sound list from a standardized and widely accessible stimulus set could promote greater consistency in procedures across studies, as researchers would be using an easily accessible stimulus set with demonstrated reliability and validity [16]. The IADS-2 are available to any Ph.D. holding faculty member by emailing the creators a request form. Replication of results could be easier if more researchers used the same standardized sound list rather than introducing variability across studies by using stimulus sets with sounds drawn from multiple sources with inconsistent sound qualities and content. The more variable and psychometrically unvalidated sounds are across studies investigating misophonia, the more difficult it may be to synthesize results and make clear conclusions about the nature, feature, and treatment of misophonia.

To address the limitations in this body of research, in the present study we used a stimulus set that has been previously validated for use in basic emotion research and standardized for length, volume, and source of audio clips. To extend research on misophonia, we examined multiple responses to standardized sounds, including affective valence, arousal, similarity to participants' triggers, and estimated avoidance of sounds. This approach was used to empirically identify standardized sounds relevant to misophonia without choosing sounds a priori.

## Current study

This study aimed to use a standardized set of sounds widely used in emotion research, the International Affective Digital Sounds-2 (IADS-2) [16], to characterize how individuals with misophonia respond to certain sounds and their associated subjective qualities (e.g., valence, arousal). We examined whether there are any IADS-2 sounds that are differentially related to misophonia symptoms, with the aim of characterizing a sound bank of misophonia triggers that is free, accessible, standardized, and has been used in basic emotion research. Investigating whether existing sounds with established norms in emotion research are associated with misophonia symptoms will allow future researchers to use a standardized list of sounds in their research and thereby increase consistency across research.

To this aim, we presented to individuals with and without misophonia a set of audio clips that were previously normed as neutral, unpleasant, and pleasant from the IADS-2. We then asked all participants to rate each sound on valence (unpleasant/pleasant), arousal (relaxing/excited), similarity to their own trigger sounds, and likelihood they would avoid these sounds. We predicted that people with high self-reported misophonia would rate typical misophonic trigger sounds (chewing, sneezing, paper rustling) as less pleasant and more arousing than people without misophonia. We also hypothesized that people with misophonia would be more likely to rate typical misophonic trigger sounds as similar to their own trigger sounds and would rate higher estimated avoidance of these sounds than those people without misophonia. Because trigger sounds may be experienced as neutral or pleasant by those without misophonia, we also predicted that misophonic ratings of positive and neutral sounds will be lower in valence and higher in arousal, similarity, and avoidance.

## Method

### Participants

Participants ($N$ = 2550) were recruited in May 2021 through CloudResearch in conjunction with Amazon's Mechanical Turk (MTurk), a crowdsourcing platform that generates data of similarly high quality to other convenience samples (e.g., college students;) [17–19]. Adhering to MTurk best practices [20], participants were required to be between 18 to 65 years of age, fluent in English, and currently residing in the United States. Participants who completed the study were compensated $7.25. Once a minimum of 225 controls were recruited, inclusion criteria were narrowed such that only participants who scored a minimum of 2 on the Misophonia Symptom Scale, a 2 on the Misophonia Emotions and Behavior Scale, and a minimum of 7 on the Severity scale of the MQ [21] could continue in the study. These cut scores were chosen because they are suggestive of significant misophonia symptomatology [21]. Participants with these scores or higher were included in the misophonia group. Maximum scores on these sections are 4, 4, and 15 respectively. At the time of study design and recruitment, similarly designed studies in misophonia research were not available to adequately conduct a power analysis. Therefore, group sizes were chosen to be adequately large while considering expense. Participants were evenly split in terms of gender across the entire sample (49.3% female, 49.3% male, and 1.3% other), averaged 38.9 years in age, and were predominantly White (see Demographics Table 1 for demographics specific to each group). There were no significant differences in demographics between the misophonia and control groups except for gender, which had a significance level of p < .001 after running a Chi-square test. In total, 2550 participants were initially recruited, with the vast majority screened out by the narrowed MQ criteria after forming the control group. A total of 377 participants remained after screening and passing attention checks, including a control group of 245 participants and a misophonia group of 132

**Table 1. Demographics and sample characteristics.**

| Variable | All | Control | Misophonia |
|---|---|---|---|
| | M(SD) | M(SD) | M(SD) |
| Age | 38.85 (10.4) | 37.51 (10.136) | 39.57 (10.49) |
| | N(%) | N(%) | N(%) |
| Male | 186 (52.5) | 141 (57.6) | 45 (34.1) |
| Female | 186 (47.5) | 103 (42) | 83 (62.9) |
| Other | 5 (1.3) | 1 (0.4) | 4 (3) |
| Race | | | |
| White | 319 (84.6) | 207 (84.5) | 112 (84.8) |
| Black or African-American | 20 (5.3) | 12 (4.9) | 8 (6.1) |
| Other Asian or Other Asian American (includes India, Malaysia, Pakistan, Philippines) | 9 (2.4) | 5 (2) | 4 (3.0) |
| More Than One Racial Group | 6 (1.6) | 3 (1.2) | 3 (2.3) |
| Korean or Korean American | 4 (1.1) | 3 (1.2) | 1 (0.8) |
| Other | 6 (1.6) | 5 (2.0) | 1 (0.8) |
| Chinese or Chinese American | 6 (1.6) | 6 (2.4) | 0 (0) |
| Japanese or Japanese American | 1 (0.3) | 1 (0.4) | 0 (0) |
| Native American, American Indian, Alaska Native | 5 (1.3) | 2 (0.8) | 3 (2.3) |
| Ethnicity | | | |
| Hispanic | 30 (8) | 21 (8.6) | 9 (6.8) |
| Non-Hispanic | 347 (92) | 224 (91.4) | 123 (93.2) |
| Education Level | | | |
| Some High School | 3 (0.8) | 1 (0.4) | 2 (1.5) |
| GED | 6 (1.6) | 4 (1.6) | 2 (1.5) |
| High School Graduate | 36 (9.5) | 23 (9.4) | 13 (9.8) |
| Bus/Tech Training Beyond High School | 9 (2.4) | 6 (2.4) | 3 (2.3) |
| Some College | 83 (22) | 53 (21.6) | 30 (22.7) |
| College Graduate | 178 (47.2) | 120 (49) | 58 (43.9) |
| Some Graduate School | 15 (4) | 8 (3.3) | 7 (5.3) |
| Masters Degree | 46 (12.2) | 30 (12.2) | 16 (12.1) |
| Doctoral Degree | 1 (0.3) | 0 (0) | 1 (0.8) |
| Household Income | | | |
| 0 - $10,000 | 23 (6.1) | 13 (5.3) | 10 (7.6) |
| $10,000 - $20,000 | 28 (7.4) | 15 (6.1) | 13 (9.8) |
| $20,001 - $40,000 | 84 (22.3) | 61 (24.9) | 23 (17.4) |
| $40,001 –$65,000 | 98 (26) | 59 (24.1) | 39 (29.5) |
| $65,001 - $100,000 | 78 (20.7) | 54 (22) | 24 (18.2) |
| More than $100,000 | 66 (17.5) | 43 (17.6) | 23 (17.4) |

participants. Participants provided implied consent (they continued onto the study procedure without signing), as data were analyzed anonymously. Participants were not warned of the potentially aversive nature of the sounds, but they were told they would be able to withdraw from the study at any time. Participants were told they would not be compensated unless the study was completed. Authors did not have access to information that could identify individuals during or after data collection. Before study procedures began, participants were given a written form describing study procedures and compensation. Participants were told that completing the study implied consent.

## Data integrity check

Several measures were taken to protect data quality. MTurk workers were only included if they historically provided high quality responses (i.e., had completed at least 1000 Human Intelligence Tasks (HITs) with an approval rate > = 99%). Participants were told in the assent that if they clicked away from the survey without completing it, they would not receive compensation. Two attention check questions were administered during the survey, and only participants who had accurate responses to both could continue the study and be used in data analyses, as accurate answers would indicate that the participants were paying attention. The first attention check ensured that participants were using browsers that supported the sound technology used in the study, and the second asked participants to demonstrate that they had heard a sample sound. This ensured that participants were both paying attention and could hear the sounds played. These measures are in accordance with data quality recommendations for MTurk-administered studies [20] and reflect measures taken in existing MTurk studies [22], as well as previous studies in our lab. Of the 2550 individuals initially recruited, 377 participants (14.8%) passed the data integrity checks, screener, completed the study, and were included in the analyses.

## Measures and materials

**International Affective Digitized Sounds (IADS-2) [16].** The IADS-2 is a standardized set of emotional sound stimuli used in research worldwide. It was validated by a normative sample and is commonly used in research of emotion and attention [23, 24]. The IADS-2 was intended to allow researchers better control over stimulus selection and to aid in research replication across labs. This sound set was developed by collecting ratings of valence, arousal, and "dominance/control" from a large sample of college students at the University of Florida (at least 100 students rated each sound). The final set of 167 sounds was chosen after three different rating studies [16].

**Misophonia Questionnaire (MQ) [21].** The MQ includes 17 items in three subscales: the Misophonia Symptom Subscale, the Misophonia Emotions and Behaviors Subscale, and the Severity Subscale. The Severity Subscale is a single item measure ranging from 1 to 15 that asks participants to indicate the extent to which sound sensitivity interferes with their lives. A score above six on the Severity Subscale indicates moderate or higher impairment related to sound sensitivities [21]. Initial validation of the MQ demonstrated good internal consistency ($\alpha$ = .86 - .89) [21]. The original study also demonstrated strong convergent and discriminant validity. Strong internal consistency was again found in a replication of the original study [25]. Cronbach's alpha for the present study was .87.

**Positive and Negative Affect Scale (PANAS)** is a widely-used scale that measures state affect. It is comprised of 20 questions, 10 of which measure positive affect and 10 of which measure negative affect [26]. Each question includes a single affect (e.g., "inspired") and participants are asked to rate how much they currently experience that affect on a 5-point Likert scale. Cronbach's alpha for the Positive Affect scale is reportedly 0.86–0.9 and Cronbach's alpha for the Negative Affect Scale is 0.84–0.87. Test-retest reliability were 0.47–0.68 and 0.39–0.71 for the positive and negative scales respectively. Cronbach's alpha in the present study for the Positive Affect Scale was .92. Cronbach's alpha in the present study for the Negative Affect Scale was .91.

**Affect Intensity Measure (AIM)** is a self-report questionnaire that uses a six-point Likert scale to measure trait-level positive and negative affect [27]. Subscales in the full measure include Negative Intensity, Positive Intensity, Negative Reactivity, and Positive Affectivity. To examine constructs associated with misophonia, the current study used only the following subscales: (1) Negative Intensity, how strongly individuals' negative emotions are in reaction to

situations (6 items), and (2) Negative Reactivity, or how easily negative emotions can be triggered by situations (6 items). Larsen and Diener [28] indicated good internal reliability for validation samples ($\alpha$ = .90-.94). Cronbach's alpha in the present study for the total scale was .90.

## Procedure

All study procedures were approved by the Duke Health Institutional Review Board and all participants provided informed consent before beginning the study. Following screening (see above), three groups of participants were each given the following instructions regarding valence and arousal ratings: "We will ask you how each sound makes you feel on a scale from 1 to 9, with 1 being 'most negative' and 9 being 'most positive.' Feel free to make ratings between these extremes as well. We will also ask you to rate the arousal of each question: this means we want to know how relaxing, soothing, or boring the sound is versus how exciting, aggravating, or intense it is. This rating is independent of valence, meaning that it doesn't matter if the arousal you feel is pleasant or unpleasant."

Each participant was then given a randomized set of 56 sounds (a random one-third of the entire set of IADS-2 for each participant to minimize participant burden associated with having to listen to all IADS-2 sounds) to rate, as well as self-report measures. Sound types (positive, negative, and neutral) were also randomized for each participant. After each sound, participants were asked to rate arousal and valence on a Self-Assessment Manikin (SAM), which is a pictorial representation of a Likert scale ranging from 1–9, where "1" refers to the lowest possible arousal or valence, and "9" refers to the highest. Following the SAM, participants are asked to rate (1) "To what extent does this sound resemble the sounds in your everyday environment that most bother you?" and (2) "To what extent would you avoid this sound in your everyday environment/ day to day life?" These questions are rated on a 1 ("not at all") to 9 ("as much as possible") Likert scale. In addition to these ratings, participants completed three measures: the MQ [21], the PANAS [26], and the AIM [27]. Participants could wait as long as they wanted after each six second sound played to move on to the ratings questions but could not replay or rewind the sound. Once they moved on to the ratings questions, they had five seconds to complete each sound rating. IADS-2 sounds are normed for volume; however, participants were able to adjust computer volume while listening.

## Data analytic plan

Group differences in sound ratings were examined using a two-way, mixed analysis of covariance (2 groups x 3 sound types, where "group" corresponds to participants with misophonia or controls, and "sound type" refers to positive, negative, or neutral sounds) on four dependent variables (ratings of valence, arousal, similarity, and avoidance). When statistically significant interactions were observed for sound type, pairwise comparisons were used to determine group differences on each dependent variable, as well as mean differences between sound type on each dependent variable.

All analyses were conducted using IBM SPSS [29] statistical software. The first step in the data analytic plan included cleaning and screening the dataset by: (a) inspecting all variables for data entry errors (none were observed), and (b) examining the normality of distributions across study variables. Next, bivariate correlations were explored to examine the relationships among variables and determine whether it would be appropriate to use any covariates. The Positive and Negative Affect Scale (PANAS), the Affect Intensity Measure (AIM), and gender were all found to significantly correlate with results and were initially included as covariates. Skewness and kurtosis levels did not exceed acceptable ranges (skewness < 2, kurtosis < 4) [30].

Finally, we ranked sounds according to a composite Z-score calculated by the sum of the Z scores from mean group differences on each sound for each dependent variable and have listed the entire ranked stimuli as "IADS-M" sounds (See Table 2). Sounds listed first (e.g., writing, whistling) reflect the sounds that most differentiate individuals with misophonia from controls. Researchers using the IADS-M can determine how many sounds to include in future studies.

## Results

To determine whether any of the responses differed as a function of sound type across groups, we conducted four univariate ANCOVAs with response as the dependent variable, Group (controls, misophonia) and Sound Type (positive, negative, neutral) as fixed factors, and gender, state affect (positive and negative), and trait affect as covariates. Across all ANCOVAs, Mauchly's Tests of Sphericity were significant (all $ps < .001$), and as such, we report the Greenhouse-Geisser corrected values for these analyses. Means for each dependent variable (valence, arousal, similarity, avoidance) as a function of Sound Type (positive, negative, neutral) and Group (controls, misophonia) are presented in Table 3.

### Valence

The first analysis, examining valence, yielded a significant main effect of Sound Type, $F(1.58, 585.12) = 7.87$, $MSE = 0.54$, $\eta_p^2 = .021$, $p = .001$, and a significant main effect of Group, $F(1, 371) = 10.07$, $MSE = 1.27$, $\eta_p^2 = .03$, $p = .002$. Finally, there was a significant Sound Type by Group interaction, $F(1.58, 585.12) = 27.11$, $MSE = 0.54$, $\eta_p^2 = .07$, $p < .001$.

To follow up on this interaction, we conducted a post-hoc analysis with a Bonferroni correction (corrected alpha .05/3 = .017) which revealed that the misophonia group rated positive sounds at a significantly lower valence (i.e., more unpleasant) than the control group ($p < .017$), with a mean difference in valence ratings of -.66 (95% CI, -.89 to -.43). The mean group difference between valence scores on negative sounds was not significant (.205; 95% CI, -.00 to .41; $p > .05$). The misophonia group rated neutral sounds at a significantly lower valence than the control group, with a mean difference in valence ratings of -.34 (95% CI, -.54 to -.14; $p < .017$).

### Arousal

The second analysis examining group differences in arousal ratings yielded a significant main effect of Sound Type, $F(1.88, 697.14) = 13.156$, $MSE = 0.50$, $\eta_p^2 = .03$, $p < .001$, and a non-significant main effect of Group, $F(1, 371) = 1.86$, $MSE = 3.83$, $\eta_p^2 = .01$, $p = .173$. Finally, there was a significant Sound Type X Group interaction, $F(1.88, 697.14) = 12.71$, $MSE = 0.50$, $\eta_p^2 = .03$, $p < .001$.

To follow up on this interaction, we conducted a post-hoc analysis with a Bonferroni correction (corrected alpha .05/3 = .017) and found that the mean group difference between arousal scores on positive sounds was not significant (.02; 95% CI, -.29 to .34; $p > .017$). The mean group difference between arousal scores on negative sounds was also not significant (.01; 95% CI, -.32 to .34; $p > .017$). However, the misophonia group rated neutral sounds at a significantly higher arousal than the control group (.56; 95% CI, .25 to .87; $p < .001$).

### Similarity to trigger sounds

The third analysis examining group differences in similarity ratings yielded a non-significant main effect of Sound Type, $F(1.48, 547.7) = 1.65$, $MSE = 1.15$, $\eta_p^2 = .00$, $p = .199$, but a

**Table 2. IADS-M sounds.**

| Sound number | Description | Type | Group Difference Composite Z-Score | Misophonia Group Valence | Control Group Valence |
|---|---|---|---|---|---|
| 358 | Writing | Neutral | -4.22 | 3.79 | 4.85 |
| 270 | Whistling | Positive | -4.10 | 3.56 | 5.27 |
| 361 | Restaurant | Neutral | -4.08 | 3.23 | 4.94 |
| 365 | Party | Positive | -3.91 | 3.98 | 5.38 |
| 724 | Chewing | Neutral | -3.91 | 2.24 | 3.88 |
| 728 | Paper 1 | Neutral | -3.72 | 3.37 | 4.52 |
| 221 | Male Laugh | Positive | -3.65 | 5.34 | 6.62 |
| 378 | Doorbell | Positive | -3.62 | 3.67 | 4.69 |
| 152 | Tropical | Neutral | -3.52 | 4.76 | 6.08 |
| 720 | Brush Teeth | Neutral | -3.52 | 3.48 | 4.59 |
| 206 | Shower | Positive | -3.48 | 4.51 | 5.6 |
| 708 | Clock | Neutral | -3.40 | 3.82 | 4.82 |
| 215 | Erotic Couple 2 | Positive | -3.33 | 4.07 | 5.65 |
| 226 | Laughing | Positive | -3.32 | 5 | 6.57 |
| 729 | Paper 2 | Neutral | -3.29 | 3.46 | 4.59 |
| 220 | Boy Laugh | Positive | -3.28 | 4.67 | 6.06 |
| 262 | Yawn | Neutral | -3.18 | 3.96 | 4.9 |
| 230 | Giggling | Positive | -3.17 | 4.47 | 5.59 |
| 704 | Phone 1 | Neutral | -3.10 | 3.13 | 4.44 |
| 355 | Crowd 4 | Positive | -3.05 | 4.53 | 5.4 |
| 723 | Radio | Neutral | -3.03 | 3.5 | 4.59 |
| 172 | Brook | Positive | -2.91 | 5.8 | 6.92 |
| 322 | Type Writer | Neutral | -2.84 | 4.14 | 4.9 |
| 352 | Sports Crowd | Positive | -2.84 | 4.74 | 5.87 |
| 374 | Sink | Neutral | -2.81 | 5.18 | 5.42 |
| 610 | Cowboy Indians | Positive | -2.77 | 3.87 | 5.04 |
| 171 | Country Night | Neutral | -2.76 | 5.28 | 5.71 |
| 252 | Male Snore | Neutral | -2.73 | 2.5 | 3.01 |
| 725 | Soda Fizz | Positive | -2.70 | 5.23 | 5.98 |
| 225 | Clap Game | Positive | -2.68 | 4.23 | 5.39 |
| 200 | Erotic Couple | Positive | -2.65 | 5 | 5.8 |
| 109 | Carousel | Positive | -2.64 | 4.28 | 5.46 |
| 368 | Crowd 5 | Neutral | -2.62 | 3.93 | 4.75 |
| 382 | Shovel | Neutral | -2.61 | 4.02 | 4.58 |
| 251 | Nose Blow | Neutral | -2.60 | 2.83 | 3.77 |
| 722 | Walking | Neutral | -2.56 | 4.67 | 4.93 |
| 700 | Toilet | Neutral | -2.56 | 4.4 | 4.98 |
| 726 | Cork Pour | Positive | -2.54 | 6.19 | 6.49 |
| 311 | Crowd 2 | Positive | -2.52 | 4.83 | 5.85 |
| 403 | Helicopter 1 | Neutral | -2.52 | 4.26 | 4.81 |
| 288 | Creep | Negative | -2.50 | 2.14 | 2.9 |
| 367 | Casino 2 | Positive | -2.50 | 5.22 | 5.93 |
| 319 | Office 2 | Negative | -2.49 | 3.09 | 3.94 |
| 102 | Cat | Neutral | -2.49 | 4.23 | 5.35 |
| 705 | Phone 2 | Neutral | -2.49 | 4.98 | 5.49 |
| 721 | Beer | Positive | -2.47 | 4.88 | 5.95 |
| 810 | Beethoven | Positive | -2.44 | 6.73 | 7.69 |

(*Continued*)

**Table 2.** (Continued)

| Sound number | Description | Type | Group Difference Composite Z-Score | Misophonia Group Valence | Control Group Valence |
|---|---|---|---|---|---|
| 110 | Baby | Positive | -2.43 | 6.21 | 6.96 |
| 709 | Alarm Clock | Negative | -2.43 | 2.57 | 3.39 |
| 320 | Office 1 | Neutral | -2.40 | 3.89 | 4.44 |
| 104 | Panting | Neutral | -2.39 | 3.11 | 3.89 |
| 216 | Erotic Couple 3 | Positive | -2.37 | 4.63 | 5.55 |
| 107 | Dog | Neutral | -2.33 | 3.85 | 5.01 |
| 703 | Busy Signal | Negative | -2.31 | 3.28 | 3.79 |
| 415 | Countdown | Positive | -2.29 | 4.7 | 4.67 |
| 373 | Paint | Neutral | -2.28 | 4.37 | 4.94 |
| 817 | Bongos | Positive | -2.25 | 5.4 | 6.71 |
| 260 | Babies Cry | Negative | -2.24 | 2.21 | 2.85 |
| 376 | Lawnmower | Neutral | -2.22 | 4.46 | 4.82 |
| 702 | Belch | Neutral | -2.20 | 2.81 | 3.27 |
| 150 | Seagull | Positive | -2.20 | 4.98 | 5.95 |
| 701 | Fan | Neutral | -2.20 | 4.8 | 4.87 |
| 353 | Baseball | Positive | -2.17 | 5.3 | 6.54 |
| 710 | Cuckoo | Neutral | -2.15 | 3.19 | 4.16 |
| 210 | Erotic Male 1 | Neutral | -2.12 | 3.17 | 4.39 |
| 826 | Bag Pipes | Positive | -2.05 | 5.16 | 5.66 |
| 360 | Roller Coaster | Positive | -2.04 | 4.05 | 5.11 |
| 730 | Glass Break | Negative | -2.04 | 2.98 | 3.35 |
| 363 | Horse Race | Positive | -2.03 | 5.37 | 5.86 |
| 611 | Battle Taps | Negative | -2.00 | 3.7 | 3.89 |
| 719 | Dentist Drill | Negative | -2.00 | 2.02 | 2.56 |
| 151 | Robin | Positive | -1.98 | 6.13 | 7.35 |
| 364 | Bar | Neutral | -1.96 | 4.21 | 4.9 |
| 224 | Kids 2 | Positive | -1.96 | 4.15 | 4.75 |
| 111 | Music Box | Positive | -1.95 | 4.27 | 4.64 |
| 310 | Crowd 1 | Neutral | -1.93 | 2.93 | 3.29 |
| 370 | Court Sport | Neutral | -1.93 | 5.4 | 5.79 |
| 242 | Female Cough | Negative | -1.93 | 2.29 | 2.94 |
| 312 | Crowd 3 | Neutral | -1.89 | 3.07 | 3.65 |
| 201 | Erotic Female 1 | Positive | -1.89 | 5.11 | 5.85 |
| 410 | Helicopter 2 | Neutral | -1.89 | 4.36 | 4.85 |
| 204 | Erotic Female 4 | Neutral | -1.87 | 3.91 | 4.75 |
| 377 | Rain 1 | Neutral | -1.82 | 5.53 | 6.33 |
| 291 | Prowler | Negative | -1.81 | 2.57 | 3.13 |
| 114 | Cattle | Neutral | -1.80 | 4.49 | 4.99 |
| 204 | Erotic Female 4 | Positive | -1.80 | 4.7 | 4.7 |
| 170 | Night | Neutral | -1.78 | 5.04 | 5.51 |
| 500 | Wind | Neutral | -1.77 | 4.71 | 4.95 |
| 245 | Hiccup | Neutral | -1.72 | 3.4 | 3.87 |
| 351 | Applause 1 | Positive | -1.71 | 5.36 | 6.43 |
| 715 | Alarm | Neutral | -1.71 | 2.95 | 3.22 |
| 130 | Pig | Neutral | -1.70 | 4.93 | 5.35 |
| 815 | Rock N Roll | Positive | -1.67 | 6.36 | 6.99 |
| 713 | Sirens | Negative | -1.66 | 3.09 | 3.4 |

(*Continued*)

**Table 2.** (Continued)

| Sound number | Description | Type | Group Difference Composite Z-Score | Misophonia Group Valence | Control Group Valence |
|---|---|---|---|---|---|
| 812 | Choir | Positive | -1.66 | 6.04 | 6.81 |
| 602 | Thunderstorm | Positive | -1.66 | 6.57 | 6.44 |
| 601 | Colonial Music | Positive | -1.63 | 5.41 | 5.95 |
| 116 | Buzzing | Negative | -1.62 | 2.15 | 3.06 |
| 717 | Slot Machine 2 | Positive | -1.60 | 5.23 | 6.44 |
| 420 | Car Horns | Negative | -1.59 | 2.39 | 2.89 |
| 295 | Couple Sobbing | Negative | -1.56 | 2.85 | 3.07 |
| 202 | Erotic Female 2 | Positive | -1.54 | 4.8 | 5.79 |
| 366 | Casino 1 | Positive | -1.52 | 5.06 | 5.82 |
| 423 | Injury | Negative | -1.45 | 2.86 | 3.13 |
| 820 | Funk Music | Positive | -1.45 | 6.47 | 6.89 |
| 808 | Bugle | Positive | -1.43 | 1.78 | 1.58 |
| 275 | Scream | Negative | -1.38 | 5.47 | 5.47 |
| 113 | Cows | Neutral | -1.37 | 5.24 | 5.47 |
| 375 | Polaroid | Positive | -1.35 | 5.4 | 6.03 |
| 716 | Slot Machine 1 | Positive | -1.34 | 6.02 | 6.67 |
| 813 | Wedding | Positive | -1.31 | 2.87 | 3.23 |
| 115 | Bees | Negative | -1.31 | 6.96 | 7.08 |
| 809 | Harp | Positive | -1.26 | 5.36 | 5.14 |
| 400 | Jet | Positive | -1.24 | 3.46 | 3.73 |
| 250 | Male Sneeze | Negative | -1.23 | 6.19 | 6.49 |
| 802 | Native Song | Positive | -1.23 | 3.98 | 4.12 |
| 243 | Couple Sneeze | Neutral | -1.19 | 2.48 | 2.7 |
| 244 | Man Wheeze | Negative | -1.18 | 4 | 4 |
| 280 | Woman Crying | Negative | -1.18 | 3.56 | 3.09 |
| 289 | Gun Shot | Negative | -1.16 | 2.57 | 2.86 |
| 241 | Male Cough | Negative | -1.14 | 3.57 | 3.61 |
| 910 | Electricity | Neutral | -1.09 | 6.84 | 87.07 |
| 816 | Guitar | Positive | -1.07 | 2.35 | 3.02 |
| 380 | Jack Hammer | Negative | -1.07 | 3.85 | 3.51 |
| 502 | Engine Failure | Negative | -1.07 | 5.16 | 5.33 |
| 425 | Train | Neutral | -1.04 | 7.04 | 7.07 |
| 811 | Bach | Positive | -1.02 | 2.9 | 2.71 |
| 293 | Man Sobbing | Negative | -0.98 | 2.8 | 2.35 |
| 296 | Women Crying | Negative | -0.97 | 5.74 | 5.44 |
| 698 | Rain 2 | Neutral | -0.95 | 3.38 | 3.48 |
| 706 | War | Neutral | -0.90 | 2.2 | 2.3 |
| 282 | Fight 2 | Negative | -0.89 | 3.02 | 3.07 |
| 283 | Fight 3 | Negative | -0.87 | 2.67 | 2.68 |
| 711 | Siren 1 | Negative | -0.85 | 2.43 | 2.42 |
| 422 | Tire Skids | Negative | -0.84 | 2.82 | 2.94 |
| 624 | AirRaid | Negative | -0.84 | 5.33 | 5.89 |
| 254 | Video Game | Positive | -0.84 | 4.05 | 4.32 |
| 133 | Growl 2 | Negative | -0.81 | 3 | 2.88 |
| 625 | May Day | Negative | -0.75 | 5.73 | 5.32 |
| 627 | Rain 1 | Neutral | -0.72 | 2.72 | 2.95 |
| 714 | Siren 2 | Negative | -0.71 | 2.6 | 2.74 |

*(Continued)*

**Table 2.** (Continued)

| Sound number | Description | Type | Group Difference Composite Z-Score | Misophonia Group Valence | Control Group Valence |
|---|---|---|---|---|---|
| 261 | Baby Cry | Negative | -0.71 | 3.64 | 3.52 |
| 626 | Explosion | Negative | -0.65 | 2.19 | 2.37 |
| 712 | Buzzer | Negative | -0.64 | 4.74 | 4.4 |
| 134 | Rattle Snake | Negative | -0.58 | 1.62 | 1.64 |
| 284 | Attack 3 | Negative | -0.58 | 3.41 | 3.24 |
| 699 | Bomb | Negative | -0.52 | 5.17 | 5.14 |
| 120 | Rooster | Neutral | -0.50 | 1.53 | 1.67 |
| 290 | Fight 1 | Negative | -0.46 | 1.74 | 1.69 |
| 276 | Female Scream 2 | Negative | -0.41 | 5.34 | 5 |
| 132 | Chickens | Neutral | -0.37 | 3.56 | 3.13 |
| 732 | Crash | Negative | -0.36 | 1.78 | 1.58 |
| 285 | Attack 2 | Negative | -0.36 | 2.09 | 1.94 |
| 292 | Male Scream | Negative | -0.21 | 3.42 | 3.42 |
| 105 | Puppy | Negative | -0.16 | 3.6 | 3.46 |
| 106 | Growl 1 | Negative | -0.15 | 1.74 | 1.65 |
| 278 | Child Abuse | Negative | -0.14 | 2.08 | 1.89 |
| 277 | Female Scream 3 | Negative | -0.12 | 2.93 | 2.38 |
| 424 | Car Wreck | Negative | 0.01 | 5.02 | 5.47 |
| 246 | Heart Beat | Neutral | 0.11 | 1.96 | 1.55 |
| 279 | Attack 1 | Negative | 0.33 | 2.8 | 2.21 |
| 600 | Bike Wreck | Negative | 0.34 | 2.36 | 2.01 |
| 281 | Attack 3 | Negative | 0.38 | 4.49 | 5.11 |
| 112 | Kids 1 | Positive | 0.45 | 2.4 | 1.69 |
| 286 | Victim | Negative | 0.78 | 2.26 | 1.69 |
| 255 | Vomit | Negative | 0.96 | 3.53 | 2.76 |
| 501 | Plane Crash | Negative | 1.03 | | |

*Note.* Ranked by decreasing difference between misophonia and control group ratings of valence

**Table 3. Descriptive properties for all ratings (N = 377).**

| Response | Sound Type | Control Mean (SD) | Skewness[1] | Kurtosis[2] | Misophonia Mean (SD) | Skewness[3] | Kurtosis[4] |
|---|---|---|---|---|---|---|---|
| Valence | Positive | 5.96 (0.96) | -.167 | 1.011 | 5.12 (1.20) | -.487 | 1.196 |
| | Negative | 2.82 (0.82) | .873 | 1.287 | 2.73 (0.94) | .743 | .572 |
| | Neutral | 4.72 (0.75) | .709 | 3.243 | 4.09 (0.97) | -.295 | 1.218 |
| Arousal | Positive | 4.15 (1.34) | -.092 | -.767 | 4.60 (1.29) | -.139 | .583 |
| | Negative | 6.00 (1.47) | -.865 | .465 | 6.53 (1.30) | -.805 | .287 |
| | Neutral | 3.98 (1.36) | -.189 | -.758 | 5.05 (1.24) | -.053 | .435 |
| Similarity | Positive | 2.33 (1.17) | 1.137 | 1.050 | 3.74 (1.51) | .451 | -.111 |
| | Negative | 4.44 (1.95) | .170 | -1.081 | 5.21 (1.74) | -.203 | -.939 |
| | Neutral | 3.11 (1.35) | .421 | -.613 | 4.70 (1.41) | .058 | -.428 |
| Avoidance | Positive | 2.66 (1.27) | .756 | -.024 | 4.24 (1.50) | .103 | -.017 |
| | Negative | 6.27 (1.69) | -.871 | .460 | 6.96 (1.41) | -1.037 | .636 |
| | Neutral | 3.61 (1.44) | .123 | -.737 | 5.33 (1.41) | -.120 | .283 |

*Note.* [1]Std. Error = .153, [2]Std. Error = .305, [3]Std. Error = .203, [4]Std. Error = .404

significant main effect of Group, $F(1, 371) = 32.17$, $MSE = 4.82$, $\eta_p^2 = .08$, $p < .001$. Finally, there was a significant Sound Type X Group interaction, $F(1.48, 547.72) = 15.13$, $MSE = 1.15$, $\eta_p^2 = .04$, $p < .001$.

To follow up on this interaction, we conducted a post-hoc analysis with a Bonferroni correction (corrected alpha .05/3 = .017) which indicated that the misophonia group rated positive sounds at a significantly higher similarity level than the control group (1.21; 95% CI, .89 to 1.52; $p < .017$). The mean group difference between similarity scores on negative sounds was not significant (.35; 95% CI, -.06 to .85; $p > .017$). The misophonia group rated neutral sounds at a significantly higher similarity level than the control group (1.18; 95% CI, .84 to 1.51; $p < .001$).

## Estimated avoidance

The fourth analysis examining group differences in avoidance ratings yielded a significant main effect of Sound Type, $F(1.71, 634.2) = 31.29$, $MSE = 0.89$, $\eta_p^2 = .08$, $p < .001$, and a significant main effect of Group, $F(1, 371) = 34.04$, $MSE = 4.33$, $\eta_p^2 = .08$, $p < .001$. Finally, there was a significant Sound Type X Group interaction, $F(1.71, 634.2) = 22.09$, $MSE = 0.89$, $\eta_p^2 = .06$, $p < .001$.

To follow up on this interaction, we conducted a post-hoc analysis with a Bonferroni correction (corrected alpha .05/3 = .017) which indicated that the misophonia group rated positive sounds at a significantly higher avoidance level than the control group (1.23; 95% CI, .91 to 1.56; $p < .001$. The mean group difference between avoidance scores on negative sounds was not significant (.29; 95% CI, -.10 to .684; $p = .14$). The misophonia group rated neutral sounds at a significantly higher avoidance level than the control group (1.18; 95% CI, .84 to 1.53; $p < .001$).

Finally, bivariate correlations to preliminarily examine the relationship between sound stimuli and misophonia symptoms (via the MQ) revealed that the MQ Symptom Subscale was significantly negatively correlated with the mean negative valence and positively correlated with arousal, similarity, and avoidance scores ($rs = -.34, .34, .21,$ and $.35$, respectively, $ps < .001$). In addition, the MQ Severity Score was positively correlated with the mean negative arousal ($r = .20$, $p = .02$) and avoidance ratings ($r = .19$, respectively, $p = .03$). A summary of these results can be found in Table 4.

Once these results were obtained, we created a list of all sounds ranked by the differences in valence ratings between the two groups (see Table 2). To do this, we took the means of the

**Table 4. Summary of results.**

| Analysis Title | Main Effects | Interaction | Post-Hoc Findings |
|---|---|---|---|
| Valence | Main Effect of Sound Type: F(1.58, 585.12) = 7.87, p = .001; Main Effect of Group: F(1, 371) = 10.07, p = .002; Interaction: F(1.58, 585.12) = 27.11, p < .001. | Post-hoc: Misophonia group rated positive and neutral sounds lower (p < .017). | Bivariate Correlations: MQ Symptom Subscale negatively correlated with mean negative valence (rs = -.34, p < .001). |
| Arousal | Main Effect of Sound Type: F(1.88, 697.14) = 13.156, p < .001; Main Effect of Group: F(1, 371) = 1.86, p = .173; Interaction: F(1.88, 697.14) = 12.71, p < .001. | Post-hoc: Misophonia group rated neutral sounds higher in arousal (p < .001). | Bivariate Correlations: MQ Symptom Subscale positively correlated with arousal (rs = .34, p < .001). |
| Similarity | Main Effect of Sound Type: F(1.48, 547.7) = 1.65, p = .199; Main Effect of Group: F(1, 371) = 32.17, p < .001; Interaction: F(1.48, 547.72) = 15.13, p < .001. | Post-hoc: Misophonia group rated positive and neutral sounds higher in similarity (p < .017; p < .001). | Bivariate Correlations: MQ Symptom Subscale positively correlated with similarity (rs = .21, p < .001). |
| Avoidance | Main Effect of Sound Type: F(1.71, 634.2) = 31.29, p < .001; Main Effect of Group: F(1, 371) = 34.04, p < .001; Interaction: F(1.71, 634.2) = 22.09, p < .001. | Post-hoc: Misophonia group rated positive and neutral sounds higher in avoidance (p < .001). | Bivariate Correlations: MQ Symptom Subscale positively correlated with avoidance (rs = .35, p < .001). |

*Note.* This table is a summary of results outlined in the Results section.

differences between scores on each sound between the two groups on all four dependent variables. We then created Z-scores for each mean (on each dependent variable). Because valence is the only scale on which participants with misophonia gave predictably lower ratings than controls, these Z-scores were reversed so that we could average the four Z-scores to create composite Z-scores. These composite Z-scores allowed us to rank each sound by how much it differentiated those with misophonia from control across the four dependent variables, with each dependent variable equally weighted (i.e., arousal, valence, similarity to one's own triggers, and estimated avoidance). Descriptive properties for the ratings are found in Table 3, while a correlation matrix of study variables is found in Table 5.

## Discussion

The primary purpose of this study was to explore whether individuals with high misophonia symptom severity differentially respond to sounds from the IADS-2 compared to healthy controls. We also aimed to curate an easily accessible sound list to be used in future misophonia research using this previously standardized and widely used stimulus set.

Several studies examining misophonia have created sound sets for idiosyncratic study use, and our study replicated these in several ways. Like previous studies [31, 32], specific cut-off scores on the MQ were used to form our misophonia group. We also used valence and arousal ratings to determine reactivity to sounds and included a similarity rating resembling that of another recent study [33]. This study also closely replicated the procedures used to validate the IADS-2; along with using the IADS-2 as our sound set, we used the SAM as our rating scale, included valence and arousal scores, and used the same rating time and duration of time between sounds. However, we extended the literature using the IADS-2 by including group comparisons on responses to the IADS-2 stimuli in both a control group and misophonia group, and by obtaining responses to stimuli regarding perceived similarity and avoidance of sounds.

Results from the present study extend previous research [6–13] by identifying a set of standardized sounds in an accessible sound bank that differentiate those with and without high misophonia symptom severity and impairment. While the IADS-2 are not completely accessible, as is the FOAMS database, it is available to any Ph.D. holding faculty who submits a request form. Unlike previous studies [6–13], the sounds in our study were standardized in terms of length, volume, duration, and other sound qualities. This ensured that inconsistencies in acoustic properties of stimuli could not confound results.

Based on these findings, we have generated a list of sounds that may be useful in future misophonia research. This stimulus set includes sounds that were not previously described in the empirical literature as the most common sounds that are aversive to those with misophonia. Accordingly, the results from this study help move the field of misophonia research closer to using standardized sounds to characterize the disorder in an empirical manner. In future studies, it will be beneficial to replicate and extend our findings by examining whether the IADS-M sounds add utility to the objective assessment and characterization of misophonia. While previous studies have demonstrated that oral-facial sounds and sounds commonly produced in offices (typing, pen clicking) are prominent triggers in misophonia [35] the top ten most differentiating sounds in the IADS-M included environmental noises, including those associated with eating or drinking (e.g., restaurant and party noises) and office sounds (e.g., paper crinkling and scribbling). This observation points to the importance of understanding the context in which triggering sounds occur, and not restricting the study of misophonia to using sounds lacking important contextual features. Indeed, several recent studies found that sounds in context, and not sounds alone, may elicit stronger aversive responses in misophonia [11, 34].

**Table 5. Correlation matrix of study variables.**

| | 1 | 2 | 3 | 4 | 5 | 6 | 7 | 8 | 9 | 10 | 11 | 12 | 13 | 14 | 15 | 16 | 17 |
|---|---|---|---|---|---|---|---|---|---|---|---|---|---|---|---|---|---|
| 1. Avg Valence Ratings for Positive Sounds | - | -.161** | -.466** | -.689** | .120* | -0.027 | -0.042 | -.168** | .587** | -.240** | -.309** | -.443** | .372** | -.281** | -.209** | -.163** | -.355** |
| 2. Avg Arousal Ratings for Positive Sounds | | - | .342** | .413** | -.311* | .658** | .273** | .344** | -.313** | .753** | .347** | .402** | .125* | .237** | .225** | 0.086 | .159** |
| 3. Avg Similarity Ratings for Positive Sounds | | | - | .756** | -.139** | .329** | .545** | .387** | -.387** | .496** | .831** | .672** | 0.076 | .273** | .354** | .194** | .500** |
| 4. Avg Avoidance Ratings for Positive Sounds | | | | - | -.222** | .338** | .273** | .501** | -.498** | .521** | .616** | .786** | -0.069 | .347** | .342** | .185** | .500** |
| 5. Avg Valence Ratings for Negative Sounds | | | | | - | -.677** | -.362** | -.703** | .551** | -.386** | -.289** | -.417** | -0.046 | 0.018 | -.255** | -.145** | -0.036 |
| 6. Avg Arousal Ratings for Negative Sounds | | | | | | - | .507** | .683** | -.380** | .760** | .461** | .505** | .217** | 0.088 | .331** | .135** | .176** |
| 7. Avg Similarity Ratings for Negative Sounds | | | | | | | - | .474** | -.239** | .429** | .679** | .414** | .227** | 0.046 | .288** | 0.085 | .210** |
| 8. Avg Avoidance Ratings for Negative Sounds | | | | | | | | - | -.426** | .462** | .490** | .664** | 0.053 | 0.081 | .293** | 0.060 | .200** |
| 9. Avg Valence Ratings Neutral Sounds | | | | | | | | | - | -.569** | -.541** | -.707** | .176** | -.208** | -.325** | -.164** | -.316** |
| 10. Avg Arousal Ratings for Neutral Sounds | | | | | | | | | | - | .638** | .710** | .124* | .266** | .375** | .158** | .363** |
| 11. Avg Similarity Ratings for Neutral Sounds | | | | | | | | | | | - | .804** | .120* | .253** | .432** | .143** | .495** |
| 12. Avg Avoidance Ratings for Neutral Sounds | | | | | | | | | | | | - | 0.005 | .307** | .439** | .144** | .498** |
| 13. PANAS Positive Subscale Scores | | | | | | | | | | | | | - | -.211** | 0.017 | 0.030 | 0.011 |
| 14. PANAS Negative Subscale Scores | | | | | | | | | | | | | | - | .389** | -0.057 | .340** |
| 15. AIM Total Score | | | | | | | | | | | | | | | - | .260** | .500** |
| 16. Participant Gender | | | | | | | | | | | | | | | | - | .237** |

(*Continued*)

**Table 5.** (Continued)

| | 1 | 2 | 3 | 4 | 5 | 6 | 7 | 8 | 9 | 10 | 11 | 12 | 13 | 14 | 15 | 16 | 17 |
|---|---|---|---|---|---|---|---|---|---|----|----|----|----|----|----|----|----|
| 17. MQ Total Scores | | | | | | | | | | | | | | | | | - |

Note

*p < .05

**p < .01.

*Note*: Ratings on each of the four dependent variables for each sound type are correlated with coviariates

*Note*: ** Correlation is significant at the 0.01 level (2-tailed); * Correlation is significant at the 0.05 level (2-tailed).

Note: 1. "Avg Valence Ratings for Positive Sounds" refers to Likert ratings from 1 to 9 on how positive or negative participants experienced the sounds played that were rated in the highest top third valenced sounds of the original manual.

2. "Avg Arousal Ratings for Positive Sounds" refers to Likert ratings from 1 to 9 on how calm or exciting participants experienced the sounds played that were rated in the top third valenced sounds of the original manual.

3. "Avg Similarity Ratings for Positive Sounds" refers to Likert ratings from 1 to 9 on how similar to bothersome sounds in their everyday environment participants experienced the sounds played that were rated in the top third valenced sounds of the original manual.

4. "Avg Avoidance Ratings for Positive Sounds" refers to Likert ratings from 1 to 9 on how much participants would avoid the sounds played in their everyday life that were rated in the top third valenced sounds of the original manual.

5. "Avg Valence Ratings for Negative Sounds" refers to Likert ratings from 1 to 9 on how positive or negative participants experienced the sounds played that were rated in the bottom third valenced sounds of the original manual.

6. "Avg Arousal Ratings for Negative Sounds" refers to Likert ratings from 1 to 9 on how relaxing or exciting participants experienced the sounds played that were rated in the bottom third valenced sounds of the original manual.

7. "Avg Similarity Ratings for Negative Sounds" refers to Likert ratings from 1 to 9 on how similar to bothersome sounds in their everyday environment participants experienced the sounds played that were rated in the bottom third valenced sounds of the original manual.

8. "Avg Avoidance Ratings for Negative Sounds" refers to Likert ratings from 1 to 9 on how much participants would avoid the sounds played in their everyday life that were rated in the bottom third valenced sounds of the original manual.

9. "Avg Valence Ratings for Neutral Sounds" refers to Likert ratings from 1 to 9 on how positive or negative participants experienced the sounds played that were rated in the middle third valenced sounds of the original manual.

10. "Avg Arousal Ratings for Neutral Sounds" refers to Likert ratings from 1 to 9 on how relaxing or exciting participants experienced the sounds played that were rated in the middle third valenced sounds of the original manual.

11. "Avg Similarity Ratings for Neutral Sounds" refers to Likert ratings from 1 to 9 on how similar to bothersome sounds in their everyday environment participants experienced the sounds played that were rated in the middle third valenced sounds of the original manual.

12. "Avg Avoidance Ratings for Neutral Sounds" refers to Likert ratings from 1 to 9 on how much participants would avoid the sounds played in their everyday life that were rated in the middle third valenced sounds of the original manual.

13. "PANAS Positive Subscale Scores" refers to ratings made of only the positive items on the PANAS.

14. "PANAS Negative Subscale Scores" refers to ratings made of only the negative items on the PANAS.

15. AIM scores reflect the strength or weakness with which individuals experience emotion, with lower scores reflecting weaker intensity of emotion, and higher scores reflecting stronger intensity.

16. Gender was coded with "0" referring to male participants and "1" referring to female participants.

Finally, we hope that this study inspires clinicians to develop innovative approaches to ethically incorporating IADS-M sounds into treatments for people with misophonia. Although most clinicians may not be able to easily access the IADS-2,), findings from this study may help clinicians assess a more heterogeneous set of possible cues beyond the oral-facial sounds consistently highlighted in the literature [35]. Sounds rarely mentioned in the literature (e.g., bird sounds) were found to be quite aversive in this study. These sounds would be unlikely to be used as trigger stimuli, as the research literature prioritizes oral-facial sounds. Again, many of the top 10 IADS-M sounds included contextual and environmental information. This may encourage clinicians to consider contextual information when assessing trigger sounds in their patients as well as understanding how trigger sounds can begin to generalize into broader

contexts (e.g., aversion to chewing sounds generalizes to aversion to restaurant sounds). This stimulus set could also be used directly in research. As one example, McKay and Frank [5] proposed the use of inhibitory learning approaches, whereby therapists use procedures to help patients inhibit unhelpful emotional behaviors in response to trigger sounds, rather than using exposures to induce habituation to sounds. Using a stimulus set as part of treatment to help develop skillful coping (cognitive, behavioral, emotional, physiological, etc.) responses to sounds may be a candidate treatment approach and aligns well with existing evidence-based transdiagnostic treatments such as the Unified Protocol [36].

We used the IADS-2 sounds to study differences in how people with high self-reported misophonia rated sounds compared to people without misophonia. We hypothesized that those with misophonia would rate neutral and positive sounds as less pleasant and more arousing than people without misophonia, because common misophonic triggers are not characterized by qualities normally aversive to the general population such as loud volume or disturbing content [37]. Further, sounds within the IADS-2 that were face-valid misophonic triggers such as chewing, scribbling, clocks ticking, and other environmental sounds were all rated as positive or neutral in the original validation studies. In line with our hypotheses, participants with misophonia rated positive and neutral sounds at a lower valence than healthy controls. However, contrary to our hypothesis, participants with misophonia did not rate positive sounds as significantly more arousing than healthy controls, yet they did rate neutral sounds as more arousing. Our hypothesized pattern of results emerged for both the similarity and avoidance scales, as participants with misophonia rated positive and neutral sounds significantly higher on both measures. There were no significant group differences for ratings on generally negative sounds. This pattern of results was unsurprising, since people with misophonia tend to be triggered by sounds that are not experienced as highly aversive to the general public. Importantly, these results were obtained after covarying for trait positive or negative affect and state negative affect (as well as gender), suggesting that individuals with misophonia react differentially to these sounds regardless of mood or a propensity to experience negative emotions.

While participants with misophonia rated positive and neutral sounds according to our hypotheses, it should not be concluded that individuals with misophonia generally like pleasant sounds less than controls. Rather, this is evidence that misophonic triggers are commonly reported as pleasant or neutral by the general population. Future studies may benefit from examining individual sounds related to misophonia irrespective of how they are categorized in the IADS-2 manual (e.g., positive, negative, neutral). Further, the discovery of trigger sounds that are normally rated as positive or neutral may also suggest that these sounds are less useful as control sounds in research unrelated to misophonia; there may be individuals with misophonia in future studies that experience these sounds as quite aversive.

This study has several limitations that should be considered and addressed in future research. A considerable limitation of this study was the reliance on self-report to define group conditions and assess dependent variables. Future studies could use clinical interviews [38], as well as more objective measures, including psychophysiology measures (e.g., galvanic skin response, neuroimaging, respiration or heart rate) to capture a more holistic picture of how individuals with misophonia respond to sounds. A multitrait-multimethod approach could strengthen and establish greater evidence of IADS-M validity, including demonstrating concurrent and discriminant validity [39]. Using psychophysiology measures would confirm self-report ratings and demonstrate concurrent validity or potentially illuminate gaps in awareness of how these individuals react to sounds. As an objective measure, our stimulus set has not demonstrated reliability and has demonstrated only preliminary construct validity in this study. Test-retest reliability is needed in future replication studies. Our results did correlate with the MQ, demonstrating concurrent validity, but we have no predictive validity since we

have not demonstrated that our results predict any real-world behavior or concurrent criterion behavior of misophonia.

Our study used cut-off scores on the MQ [21] to form our misophonia group, which precluded a dimensional analysis of misophonia. Future studies might consider using dimensional assessments and analyses, so that a broader proportion of sufferers can be studied and so that we can more dimensionally understand the relationship between sound ratings and severity. Future studies should consider using a clinical control group with similar features to misophonia (e.g., high emotion dysregulation, hyperacusis, perfectionism, etc.) to determine whether features other than sound tolerance may be driving these results or whether this pattern of results is unique to misophonia. Dimensional studies would also help elucidate sound sensitivities that may exist in the control group, as sound sensitivity or aversion to specific sounds is not specific to misophonia. Alternatively, researchers using auditory stimuli may want to screen for misophonia, as these participants may provide unexpected results.

Finally, the diversity of participants in this study was not proportional to the racial breakdown of the United States, as the sample was primarily White. Therefore, we are missing data on how people from different races or cultures react to these sounds, and therefore may not have developed a fully generalizable sound set. Future studies should recruit a more diverse pool of participants to increase ecological validity and accurately represent the population of individuals with misophonia.

## Author Contributions

**Conceptualization:** Jacqueline Trumbull, M. Zachary Rosenthal.

**Data curation:** Jacqueline Trumbull.

**Formal analysis:** Jacqueline Trumbull, Noah Lanier, Katherine McMahon, Rachel Guetta.

**Methodology:** Jacqueline Trumbull.

**Project administration:** Jacqueline Trumbull.

**Supervision:** M. Zachary Rosenthal.

**Validation:** Jacqueline Trumbull.

**Writing – original draft:** Jacqueline Trumbull.

**Writing – review & editing:** Jacqueline Trumbull, Noah Lanier, Katherine McMahon, Rachel Guetta, M. Zachary Rosenthal.

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
