## [Decision Letter · Decision Letter 0]

6 Sep 2023

PONE-D-23-17567Using a Standardized Sound Set to Help Characterize Misophonia: The International Affective Digitized SoundsPLOS ONE

Dear Dr. Trumbull,

Thank you for submitting your manuscript to PLOS ONE. After careful consideration, we feel that it has merit but does not fully meet PLOS ONE’s publication criteria as it currently stands. Therefore, we invite you to submit a revised version of the manuscript that addresses the points raised during the review process. 1. Both reviewers suggested that the way in which the results were presented was at times confusing and could benefit from substantial revision and, in some cases, additional or revised summary figures/tables.2. Both reviewers raised questions about the statistical models used. In particular, the first reviewer questioned whether mixed-effect ANOVAs might be more appropriate than ANCOVAs depending on which covariates were included given that nothing about the possible covariates was reported.3. Both reviewers suggested including more information about how control subjects were recruited.4. The second reviewer, who included their name in the review, flagged for the authors the existence of a paper that seems quite relevant for the present study. Please consider referencing this work.5. The authors should include in their revision a link to an open-access repository where readers can access the data (e.g. osf.io).

We look forward to receiving your revised manuscript.

Kind regards,

Andrew R Dykstra

Academic Editor

PLOS ONE

Journal Requirements:

Reviewers' comments:

Reviewer's Responses to Questions

**Comments to the Author**

1. Is the manuscript technically sound, and do the data support the conclusions?

Reviewer #1: Yes

Reviewer #2: Yes

2. Has the statistical analysis been performed appropriately and rigorously? 

Reviewer #1: I Don't Know

Reviewer #2: Yes

3. Have the authors made all data underlying the findings in their manuscript fully available?

Reviewer #1: No

Reviewer #2: Yes

4. Is the manuscript presented in an intelligible fashion and written in standard English?

Reviewer #1: Yes

Reviewer #2: Yes

5. Review Comments to the Author

Reviewer #1: The present paper reports one study aimed at developing a standardized stimulus set for use in misophonia research. On-line participants were presented with a large set of stimuli from an already published and standardized corpus of sounds, the International Affective Digitized Sounds (IADS). Participants with and without misophonia provided valence and arousal ratings, plus ratings of similarity to one’s own triggers and likelihood of stimulus avoidance. These data allowed them to rank the stimuli based on the extent to which those stimuli elicited distinct responses from those who report higher levels of misophonia symptom severity.

The work is interesting and useful because, as the authors note, most instruments focus on self-report without actually measuring how listeners respond to various sounds, and among the studies that do present sounds to listeners, stimulus sets vary widely. It’s potentially very useful for both clinical and basic science to better understand how individuals who do and do not suffer from the disorder of misophonia experience react to real world sounds so that responses to the same stimuli can be compared across studies. I think this is a potentially valuable contribution, however some of the results and methodological choices could be presented more clearly.

The bulk of the results section reports a series of mixed design ANCOVAs on the four dependent measures with the between-subjects variable group and within-subjects variable of sound type—but were the covariates of PANAS and AIM scores also included in these analyses? If so, nothing is reported about these covariates. If not, then why not report mixed design ANOVAs? Just for clarity, the structure of each section reporting ANCOVA/ANOVA results could be a bit more straight-forward. Ideally, the authors would first report and then describe main effects --with overall means for each level of the variable-- and then report and describe the interactions and post hoc analyses with means (instead of reporting main effects and interactions, then describing the interactions, and then only describing one of the main effects and not reporting means). The overall means for the two groups and the three sound types are only reported as difference scores (which are a bit hard to interpret at times), and it would be much clearer to report means either in the text itself or at least in the table.

A minor comment regarding the analysis on similarity ratings: it is puzzling that the main effect of sound type was not significant yet the post hoc comparisons were significant? First, usually post hoc tests are only run when the main effect (or interaction) is significant, so it’s not clear why those analyses were conducted in the first place. But in addition, why would post hoc tests be significant when the main effect was not? Is this a mistake?

The ranking of the stimuli in Table X/Table 2 is very confusing and the figure caption is lacking important information. What are the two columns labeled “Valence”? Reading down from Table 1 the reader might assume that the left column is controls and the right misophonia group, however the values probably reflect group differences in ratings for each stimulus since the rankings are based on this (but which difference scores for which variables are shown? Just valence?). This table could be far more useful and interesting if means for each rating were provided alongside the composite group difference Z-scores for each stimulus. The initial description of the Z-scores and reference to this table was also very confusing and should be simplified (see comment below).

How was the control group selected? Was there an inclusion threshold, or was this group made up of anyone who did not exceed the inclusion threshold for the misophonia group? Was any comparison made between data obtained from prior research (e.g. valence and arousal ratings) and the control group in the present study? It would be reassuring to see that the control group’s ratings are similar to ratings obtained from typical listeners in past studies/standardizations, if possible. For me this raises a bigger question about the assumption that “misophonia" is only experienced by those who score high on instrument like the MQ. While misophonia can be defined as a disorder, it is also clear that many people experience strong aversive emotional (“misophonic”) responses to trigger sounds but do not necessarily experience the life disruptions and impairments of the disorder misophonia. Do we assume that these aversive responses are qualitatively different from those experienced by the misophonia group? The present data set seems like it has the potential to address some of these questions by looking at heterogeneity within the control group. If that is beyond the scope of the present work, then it would at least be worth discussing somewhere in the paper and making all of the data available to other researchers who might be interested in this question.

Minor comments:

The document has no page numbers. Reviewing is easier when everyone can refer to page numbers that are inserted in the document (as opposed to relying on page numbers in a particular pdf reader, which may be different). The pages mentioned below correspond to the page numbers in the pdf when I open it in Adobe Reader.

The PANAS and AIM were administered and included in the analyses, but no results were reported other than the correlation table which is not referenced in the text at all. This table also lists variables that are not described adequately in the text or figure caption (how is gender coded? What does a higher AIM score mean?).

Methods: All participants were asked to rate the similarity between each stimulus and sound triggers in their own environment, but this question seems like it would make a lot more sense to someone with misophonia than with someone who does not experience misophonia or is not aware of misophonia. Were the listeners given any particular instructions to help them interpret this question?

p. 13, last paragraph before results section: the levels of the variables should be explicitly described (e.g. “sound types” corresponds to positive, negative, neutral and “group” corresponds to misophonia and controls). Also, the final sentence before the results section is extremely confusing and should be revised for simplicity (e.g. “we ranked sounds based on the relative similarity of ratings provided by those with and without misophonia (see below for detailed explanation)”).

“Table X” is referenced throughout the paper-- presumably this is Table 2?

Reviewer #2: [Note: Please see attachment for a formatted version of this review]

This paper makes a compelling case for the need for standardized stimuli in misophonia research and offers a unique solution to that problem. The authors presented controls and misophonic individuals with sounds from IADS-2, a standardized set of emotional sound stimuli, and noted significant differences in how misophonic individuals viewed canonically positive and neutral sounds compared to controls. The authors rank the list of sounds by how large the difference in ratings is between misophonic individuals and controls, and subsequently propose the top 10 differentiating stimuli as “IADS-M” sounds for use in further research/therapy. This work contains some important points and interesting findings, but some restructuring and additional detail is necessary before it is suitable for publication.

Before listing specific comments, I want to disclaim my involvement in and draw the authors’ attention to the Free Open-Access Misophonia Stimuli (FOAMS) database, recently published in the Journal of Open Psychology Data, of which I am an author. This project similarly identifies the gap in misophonia literature that the authors discuss in the present manuscript – namely the lack of (and need for) a standardized set of misophonic stimuli – but attempts to bridge that gap in an orthogonal way. It is my opinion that the present manuscript still holds value and ought to be published. However, the authors should consider referencing FOAMS and/or differentiating their study from what FOAMS provides. I fully acknowledge that FOAMS may not have been available when the authors were preparing this manuscript (the FOAMS database was first posted online September 2022, and not published until August 2023), but the FOAMS paper may help emphasize the authors’ points. I am comfortable deferring to the handling editor’s judgment as to whether including FOAMS is appropriate.

Major comments:

1. Arguments in the manuscript would benefit from reframing:

1.1. Use of self-report:

Introduction paragraph 2 knocks the reliance on self-report measurements – for a variety of valid reasons – and advocates for more objective measures. However, the paradigm the authors employ is entirely self-report, and they discuss this as being a limitation in the Discussion. Thus, this argument in the introduction should be omitted/restructured.

While asking participants to rate the sound of chewing is substantially better than asking them in text how they feel about chewing, it still suffers from response biases, etc. For example, the participants are asked to rate “arousal” on a scale from “calm” to “excited”, and I am interested in how much instruction/coaching was given to understand that question. (Anecdotally, I’ve struggled in my own work with getting participants to think of arousal non-sexually, and I personally don’t know if I’d rate even my most triggering sounds as “exciting”). Further, while the Similarity and Avoidance questions are interesting data points, I’m not convinced they accurately capture whether or not a misophonic person is triggered by that stimulus. (For instance, if I’m triggered by chewing sounds but the particular IADS-2 chewing stimulus doesn’t land for me, how would I answer “To what extent does this sound resemble the sounds in your everyday environment that most bother you?”)

Perhaps, instead, the authors could comment on the wide array of self-reported prompts being asked in misophonia literature (e.g., discomfort, unpleasantness, tolerability, distress, annoyance, etc.) and highlight the utility of asking more standardized questions like valence/arousal, as doing so might help contextualize misophonia in relation to other auditory/psychiatric disorders that have data on valence/arousal.

1.2. Positive/Neutral/Negative sounds:

The use of these terms throughout the manuscript would benefit from some clarity/discussion. The key finding presented in the abstract is that misophonic participants rate pleasant and neutral sounds more negatively than controls do, which I think evokes a different image than what the study actually shows. Please correct me if I’m wrong, but it’s not so much that misophonic individuals hate classically pleasant sounds (which would be an interesting – but different – discussion), but that whoever provided normed ratings for IADS classified sounds as “positive” that are coincidentally also everyday sounds commonly reported as misophonic triggers. After staring at Table 2 for a while, my major takeaway is that what is positive vs. neutral vs. negative is super subjective (and probably differs by lived experience), and that there appears to be high heterogeneity within each category.

Is the primary goal of this paper to compare ratings for sounds categorized as positive vs. neutral vs. negative, or to provide a list of sounds that differentiate misophonics from controls? If the former, I might suggest digging into the IADS-2 normed ratings a bit more and re-categorizing the sounds post hoc (e.g., highly positive, positive, positive-neutral, negative-neutral, negative, highly negative) so each category is a bit more homogenous. Alternatively, you could apply your own higher-level categories labels (e.g., “office”, “animal”, etc.), which I mention below (comment 3.2), and make a more specific content claim. However, if your main goal is to introduce the IADS-M, consider focusing less on the positive vs. neutral vs. negative differences and more on the stimuli as individual units.

1.2.1. Sidenote: Perhaps these data also speak well to another issue I’ve struggled with articulating in misophonia research, which is “what sounds are truly neutral?!” These data demonstrate that sounds normed as Neutral (e.g., Brush Teeth, Clock) can also be bothersome to individuals with misophonia, so the authors might consider adding a word of caution against incorporating them as “control” sounds in future research…

1.3. Utility:

I’d suggest thinking more about who the audience of the IADS-M is. I discuss this in more detail below (comment 2.1.1), but the IADS-2 isn’t entirely open-access (especially in relation to something like FOAMS), so it might be less feasible for new researchers to incorporate these sounds into their research. Perhaps, instead, it’s a call to researchers or clinicians(?) who presently use the IADS-2 in auditory research to additionally screen for misophonia in their samples. Maybe the IADS-M can be a useful word of caution to researchers using IADS, that if you use those 10 sounds, you might inadvertently get unexpected results if your participants have misophonia. Regardless, Introduction and Discussion sections could be strengthened with a clearer focus of the utility of the IADS-M.

2. The manuscript is missing key information:

2.1. IADS-2

2.1.1. Accessibility: Throughout the manuscript, IADS-2 is referred to as “easily accessible.” The authors should include the steps required to obtain IADS-2, as not everyone is eligible to request them (and the process isn’t immediate), and consider caveating claims of accessibility. For instance, the IADS-2 request form requires that the “Requestor must be a PhD-holding faculty at a non-profit, degree-granting, academic institution,” with a “direct link to their official faculty profile on their institutional website that lists their credentials as a PhD-holding faculty”. Thus, I am unclear whether clinicians could actually utilize IADS-M sounds in their treatments, unless they have connection to a PhD-holding faculty member.

2.1.2. Stimulus characteristics: The Method section should include more information regarding the content, length, volume, etc. of the IADS-2 sounds and the task paradigm (especially since Discussion paragraph 2 mentions “…the sounds in our study were standardized in terms of length, volume, duration, and other sound qualities”). Some questions I had include:

2.1.2.1. Did sounds play automatically, or were they controlled by participants?

2.1.2.2. Was the length of sound presentation sufficient to identify what the sound was? (cite previous misophonia research)

2.1.2.3. Was there any time limit to submit ratings?

2.1.2.4. What was the inter-stimulus interval?

2.1.2.5. Were positive/neutral/negative sounds blocked or intermixed?

2.1.2.6. How were the set of 56 sounds determined for each participant? (e.g., did they contain equal amounts of positive/neutral/negative stimuli?)

2.1.2.7. How was volume standardized across participants? Did they wear headphones, or play the sounds aloud? Could participants have muted/changed volume midway through the experiment?

2.1.2.8. Did participants know in advance that they would be hearing erotic/aversive stimuli? Could participants withdraw if they were uncomfortable?

2.1.3. IADS-M: Unless I missed this, were the 10 IADS-M sounds ever mentioned/listed anywhere? Consider discussing them in text (e.g., Results, last paragraph) and/or denoting them in Table 2. Further, why 10?

2.2. Participants:

2.2.1. More detail on group inclusion criteria would be useful. I’m less familiar with the MQ, but where do the cutoffs of 2 on the Symptom and Emotions/Behavior Scales come from? It initially reads like a very low cutoff – perhaps include the max value that the scales are out of for context.

2.2.2. Why was the control group recruited before the misophonic group? Do the samples differ unintentionally due to the passage of time?

2.2.3. If inclusion criteria narrowed after 225 controls were recruited, why did the final control group include 245 participants? What determined the 225 (initial) stopping number – was there an a priori power analysis?

2.2.4. Why were there unequal numbers of participants in the control vs. misophonia groups? While equal gender splits across the entire sample is impressive, looking at Table 1, it does not hold true within sample; did the groups differ significantly in terms of gender, age, etc.?

2.2.5. Starting with 2550 participants and ending with 377 seems like a huge drop off. Can you comment on why? What proportion of the excluded participants failed the screening vs. attention checks?

2.2.6. How much were participants compensated to participate?

2.2.7. Ethics: I trust that if the study procedures were approved by the university IRB then this is fine, but the current phrasing of the ethics statements (e.g., “implied consent”) read extremely sketchy. Consent should be obvious, not implied. As above (comment 2.1.2.8), were participants informed about the potentially aversive nature of the study? Could they withdraw without penalty?

3. The manuscript would benefit from additional analyses/visuals:

3.1. Additional analyses would be helpful to support the idea that IADS-M sounds are useful for differentiating individuals with misophonia. For instance, does a classifier built on responses to these sounds significantly predict group membership? Do the ratings of valence/arousal/etc. of these sounds in particular correlate the strongest with MQ score?

3.2. Consider additional category breakdowns (e.g., “office” sounds, as referenced in Discussion), instead of just positive/negative/neutral. Are the results that individuals with misophonia respond with lower valence to normed positive stimuli driven by any category in particular? Doing so might illuminate a more nuanced understanding of misophonic triggers.

3.3. Consider adding summary figures to visually parse the results, as otherwise the Results section text is dense. Additionally, unless you have specific predictions about differences between sound types collapsed across group membership, consider omitting these paragraphs (i.e., Results paragraphs 5, 8, etc.) from the Results section since they aren’t really discussed further. Perhaps they could be moved to a table or Supplement.

3.4. Clarify statistics/multiple comparison corrections:

3.4.1. I am happy to see the authors were conscious of corrections and included a Bonferroni-corrected alpha in Results paragraph 4. However, it is unclear how often/which results were corrected (e.g., is Results paragraph 7 corrected? Paragraph 10?), since the post hoc stats are still reported at an alpha of 0.05.

3.4.2. Table 4: Are these correlations corrected for multiple comparisons?

3.4.3. Results, paragraph 1: Were correlations run for each subscale of the MQ, or only Total scores? If the former, consider adding the individual subscale correlations to Table 4, so results reported in text can be visualized in context of other variables.

Minor comments:

4. Needs more citations:

4.1. Introduction, paragraph 8: Throughout

4.2. Method, Measures and Materials: Provide example studies for “commonly used in research of emotion and attention”

4.3. Discussion, paragraph 2: Provide examples of “Like previous studies,”

4.4. Discussion, paragraph 3: Provide examples of “extend previous research” and “Unlike previous studies”

4.5. Discussion, paragraph 4: Throughout. Link your findings of environmental/office sounds (or non-oral/nasal sounds in general) being bothersome with previous literature that reports similar results.

4.6. Discussion, paragraph 5: Throughout. Also, for what it’s worth, Hansen et al. (2021) does use bird sounds as a stimulus, although it wasn’t found to be tremendously bothersome (to either individuals with misophonia or controls).

4.7. Throughout: consider referencing FOAMS

5. Line edits:

5.1. Abstract: Add a comma after “(IADS-2)”, or remove the comma after “Bradley and Lang”

5.2. Introduction, paragraph 1: I’d recommend adding “in 2013” somewhere in the last sentence to contextualize Schroder et al.’s work and clarify the timeline of misophonia research

5.3. Introduction, paragraph 3: Start a new paragraph at “In contrast, if a more standardized approach…”

5.4. Introduction, paragraph 5: I’d recommend reframing this paragraph, as “most studies have been limited to study team expertise and a review of the literature” feels like an attack on otherwise good practices. I think your underlying point is valid and important – that if we don’t branch out and try other sounds, we may be missing out on other crucial trigger sounds that aren’t as commonly reported – but it could be made in a different way. For example, I’ve gone for something like “participants report a wide variety of trigger sounds, but only a small subset of sounds is actually used in studies”. Also, I’d recommend removing “the tail wagging the proverbial dog”, as it feels a little too colloquial.

5.5. Introduction, last paragraph: Rephrase “estimated avoidance of sounds avoid them”

5.6. Current Study, last paragraph: Add “people” to “…avoidance of these sounds than those [people] without misophonia” (I initially read it as “…these sounds than those sounds…”)

5.7. Method, Participants: Change “data was” to “data were”

5.8. Method, Data Integrity Check: Consider renaming “attention check” (“technology check”?), since the two questions listed don’t explicitly measure participants’ ability to direct or maintain attention.

5.9. Method, Measures and Materials: As with MQ, introduce acronyms in bold following the name of the measure at the start of each paragraph. Then, just refer to them all with acronyms in the last sentence of Method, Procedure.

5.10. Method, Measures and Materials, Positive and Negative Affect Scale: Rephrase the sentence “Cronbach’s alpha for the Positive Affect scale is reportedly 0.86-0.9 and 0.84-0.87 for the Negative Affect Scale” with parallel structure

5.11. Method, Procedure: Add “as” to “…as well [as] self-report measures”

5.12. Method, Procedure: Give anchors/more detail to “a Likert scale ranging from 1-9”. (e.g., is 1 low or high arousal/valence?)

5.13. Method, Data Analytic Plan: Swap the order of the first 2 paragraphs (otherwise “these analyses” and “all four ANCOVAS” are confusing without explanation)

5.14. Method, Data Analytic Plan (and elsewhere): Replace “Table X”

5.15. Results, paragraph 1: “the MQ Symptom Subscale was significantly negatively correlated with the mean negative valence, arousal, similarity, and avoidance scores (rs = -.34 .34, .21, and .35,…” – was MQ negatively correlated with all four measures, or just the first? Rephrase the text or the r-values accordingly.

5.16. Results, paragraph 8: Remind the reader why p = 0.024 is not significant (i.e., that corrected alpha is 0.017)

5.17. Results: For stats in which both mean ratings and CI are reported in parentheses, keep consistent where parentheses are reported -- e.g., (.205 (95% CI, -.00 to .41; p > .05) is missing a parenthesis

5.18. Results: Replace all instances of “p = 0.000” with “p < 0.001”

5.19. Results: Keep reporting of p-values consistent. E.g., sometimes exact values are reported (e.g., p=.024, p=.173), sometimes approximation (e.g., p<.05, p>0.05)

5.20. Discussion, paragraph 5: Rephrase “These sounds were not obviously triggering, as they were not oral-facial…” At face value, I think it’s been well-established that non-orofacial sounds can also be triggering; if this was your point, I’d remove this sentence. However, if your point is that this type of sound wouldn’t classically be used as a “trigger” stimulus because it’s not orofacial and most literature still prioritizes orofacial sounds, I think that’s an important discussion point and should be rephrased for clarity.

5.21. Discussion, last paragraph: Since your research question doesn’t involve development or measurements over time, cross-sectional vs. longitudinal design limitations do not feel necessary. I’d recommend removing the first bit and starting the paragraph at “Finally, the diversity of participants…”

5.22. Table 1: Report a consistent number of decimal spots for all data values. Also, is the percentage for White individuals with Misophonia cut off?

5.23. Table 2: Why are there two columns for Valence?

5.24. Table 4: Keep the anchor words for the ratings consistent throughout the manuscript (e.g., the text says “calm/excited” but this caption says “relaxing or exciting”).

5.25. Table 4: Change “7. Ave Similarity…” to “7. Avg Similarity”. Also, the caption to Table 4 would be easier to parse with the sentences beginning “Avg Ratings for…” numbered as in the table and split on separate lines.

Overall, thanks for the great work! I look forward to seeing an updated version.

Signed,

Heather Hansen

6. PLOS authors have the option to publish the peer review history of their article (what does this mean?). If published, this will include your full peer review and any attached files.

Reviewer #1: No

Reviewer #2: **Yes: **Heather Hansen

---

## [Author Response · Author response to Decision Letter 0]

1 Feb 2024

RESPONSE TO REVIEWERS

Response: We appreciate the editors and reviewers time and consideration of our manuscript. We believe our responses below adequately address the important concerns and feedback noted. Overall, we believe the changes made have significantly strengthened the manuscript and appreciate your consideration of the revised manuscript for publication in the Journal of Traumatic Stress.

REVIEWER 1

The present paper reports one study aimed at developing a standardized stimulus set for use in misophonia research. On-line participants were presented with a large set of stimuli from an already published and standardized corpus of sounds, the International Affective Digitized Sounds (IADS). Participants with and without misophonia provided valence and arousal ratings, plus ratings of similarity to one’s own triggers and likelihood of stimulus avoidance. These data allowed them to rank the stimuli based on the extent to which those stimuli elicited distinct responses from those who report higher levels of misophonia symptom severity.

The work is interesting and useful because, as the authors note, most instruments focus on self-report without actually measuring how listeners respond to various sounds, and among the studies that do present sounds to listeners, stimulus sets vary widely. It’s potentially very useful for both clinical and basic science to better understand how individuals who do and do not suffer from the disorder of misophonia experience react to real world sounds so that responses to the same stimuli can be compared across studies. I think this is a potentially valuable contribution, however some of the results and methodological choices could be presented more clearly

Comment 1: 

The bulk of the results section reports a series of mixed design ANCOVAs on the four dependent measures with the between-subjects variable group and within-subjects variable of sound type—but were the covariates of PANAS and AIM scores also included in these analyses? If so, nothing is reported about these covariates. If not, then why not report mixed design ANOVAs? Just for clarity, the structure of each section reporting ANCOVA/ANOVA results could be a bit more straight-forward. Ideally, the authors would first report and then describe main effects --with overall means for each level of the variable-- and then report and describe the interactions and post hoc analyses with means (instead of reporting main effects and interactions, then describing the interactions, and then only describing one of the main effects and not reporting means). The overall means for the two groups and the three sound types are only reported as difference scores (which are a bit hard to interpret at times), and it would be much clearer to report means either in the text itself or at least in the table.

Response 1: Regarding the reporting of results, we are under the impression that one always reports whether the interaction is significant or not and then reports the results from the main effects only if the interaction is not significant. This impression is derived from the following source: As a general rule, you do not report main effects when there is a statistically significant two-way interaction effect (Maxwell & Delaney, 2004). " We hope that clarifies why we reported results that way. We have edited the paper to remove reference to analyses done without covariates and have only reported results the come from analyses with covariates.

Comment 2:

A minor comment regarding the analysis on similarity ratings: it is puzzling that the main effect of sound type was not significant yet the post hoc comparisons were significant? First, usually post hoc tests are only run when the main effect (or interaction) is significant, so it’s not clear why those analyses were conducted in the first place. But in addition, why would post hoc tests be significant when the main effect was not? Is this a mistake?

Response 2: We acknowledge the reviewer's concern. The decision to run post-hoc tests was based on the significant interaction observed in the data. The interaction effect, specifically between Sound Type and Group, was found to be significant (p<.001), prompting further analysis despite the non-significant main effect. Also, when an interaction effect is significant, it suggests a more complex relationship between variables that can't be fully understood just by looking at main effects. Post-hoc tests following a significant interaction effect can show how and where these interactions occur between the levels of the independent variables.

Comment 3:

The ranking of the stimuli in Table X/Table 2 is very confusing and the figure caption is lacking important information. What are the two columns labeled “Valence”? Reading down from Table 1 the reader might assume that the left column is controls and the right misophonia group, however the values probably reflect group differences in ratings for each stimulus since the rankings are based on this (but which difference scores for which variables are shown? Just valence?). This table could be far more useful and interesting if means for each rating were provided alongside the composite group difference Z-scores for each stimulus. The initial description of the Z-scores and reference to this table was also very confusing and should be simplified (see comment below).

Response 3: We agree with you that more information should be provided in the tables. We have now included two more tables that show sound ratings for both groups across all ratings and have clarified the valence score in Table 2.

Comment 4:

How was the control group selected? Was there an inclusion threshold, or was this group made up of anyone who did not exceed the inclusion threshold for the misophonia group? Was any comparison made between data obtained from prior research (e.g. valence and arousal ratings) and the control group in the present study? It would be reassuring to see that the control group’s ratings are similar to ratings obtained from typical listeners in past studies/standardizations, if possible. For me this raises a bigger question about the assumption that “misophonia" is only experienced by those who score high on instrument like the MQ. While misophonia can be defined as a disorder, it is also clear that many people experience strong aversive emotional (“misophonic”) responses to trigger sounds but do not necessarily experience the life disruptions and impairments of the disorder misophonia. Do we assume that these aversive responses are qualitatively different from those experienced by the misophonia group? The present data set seems like it has the potential to address some of these questions by looking at heterogeneity within the control group. If that is beyond the scope of the present work, then it would at least be worth discussing somewhere in the paper and making all of the data available to other researchers who might be interested in this question.

Response 4: 

Thank you for requesting this clarification. We have included new tables that show our misophonia and control group’s sound ratings. Our control group included anyone that did not meet the MQ threshold outlined in the paper, which brings up your interesting observation about control group variability. Our team has discussed this in great detail. We believe that our sample that reflects two groups of people that are significantly different in misophonia severity, as evidenced by differences in mean scores. However, we also agree with you that in the control group, there is variability in misophonia symptoms. And we further agree that aversive responses to sounds is not specific to misophonia and is experienced by many people who do not meet symptom and severity levels to meet for misophonia. In our discussion, we highlight how future studies could more thoroughly examine variability in misophonic symptoms vis a vis symptoms of sound aversion or sound intolerance that are not misophonia per se and have added that there are likely misophonic symptoms or sound aversions within the control group. We do agree with the reviewer that this was not the primary purpose of the study and therefore have elected not to include supplementary analyses addressing control group variability. We point the reviewer to a recent study on misophonia and sensory intolerance that discuss this issue (Jakubovski, E., Müller, A., Kley, H., de Zwaan, M., & Müller-Vahl, K. (2022). Prevalence and clinical correlates of misophonia symptoms in the general population of Germany. Frontiers in psychiatry, 13, 1012424.)

Minor Comments

Comment 5:

The document has no page numbers. Reviewing is easier when everyone can refer to page numbers that are inserted in the document (as opposed to relying on page numbers in a particular pdf reader, which may be different). The pages mentioned below correspond to the page numbers in the pdf when I open it in Adobe Reader.

Response 5: Thank you for pointing out this mistake. The document absolutely needs page numbers, so those have been added. 

Comment 6:

The PANAS and AIM were administered and included in the analyses, but no results were reported other than the correlation table which is not referenced in the text at all. This table also lists variables that are not described adequately in the text or figure caption (how is gender coded? What does a higher AIM score mean?).

Response 6: I appreciate your request for greater clarification in the table, which has been edited to reflect your comments. We report in the discussion that the AIM and PANAS were included as covariates but did not significantly change the pattern of results. However, we did not explicitly name that the PANAS and AIM scores were used as covariates, which understandably leads to confusion; this is now explained in the data analytic plan on page 7. 

Comment 7:

Methods: All participants were asked to rate the similarity between each stimulus and sound triggers in their own environment, but this question seems like it would make a lot more sense to someone with misophonia than with someone who does not experience misophonia or is not aware of misophonia. Were the listeners given any particular instructions to help them interpret this question?

Response 7: We appreciate you pointing out that individuals without misophonia may be confused by the similarity question if they do not consider any sounds to be triggering. We have now added the instructions given to participants to the Method section. 

Comment 8:

p. 13, last paragraph before results section: the levels of the variables should be explicitly described (e.g. “sound types” corresponds to positive, negative, neutral and “group” corresponds to misophonia and controls). Also, the final sentence before the results section is extremely confusing and should be revised for simplicity (e.g. “we ranked sounds based on the relative similarity of ratings provided by those with and without misophonia (see below for detailed explanation)”).

Response 8: Thank you for this comment and helping to bring greater clarity to this section. Revisions reflecting your suggestions have been added to the data analytic plan.

Comment 9:

“Table X” is referenced throughout the paper-- presumably this is Table 2?

Reviewer #2: [Note: Please see attachment for a formatted version of this review]

Response 9: Thank you for pointing out this mistake. Table X does refer to Table 2 and the manuscript has been revised accordingly.

Comment 10

This paper makes a compelling case for the need for standardized stimuli in misophonia research and offers a unique solution to that problem. The authors presented controls and misophonic individuals with sounds from IADS-2, a standardized set of emotional sound stimuli, and noted significant differences in how misophonic individuals viewed canonically positive and neutral sounds compared to controls. The authors rank the list of sounds by how large the difference in ratings is between misophonic individuals and controls, and subsequently propose the top 10 differentiating stimuli as “IADS-M” sounds for use in further research/therapy. This work contains some important points and interesting findings, but some restructuring and additional detail is necessary before it is suitable for publication.

Response 10: Thank you for these revisions which have added clarity for readers and strengthened the manuscript. We really appreciate the time you spent reviewing our study, and our manuscript is greatly improved by your comments.

Reviewer 2

Before listing specific comments, I want to disclaim my involvement in and draw the authors’ attention to the Free Open-Access Misophonia Stimuli (FOAMS) database, recently published in the Journal of Open Psychology Data, of which I am an author. This project similarly identifies the gap in misophonia literature that the authors discuss in the present manuscript – namely the lack of (and need for) a standardized set of misophonic stimuli – but attempts to bridge that gap in an orthogonal way. It is my opinion that the present manuscript still holds value and ought to be published. However, the authors should consider referencing FOAMS and/or differentiating their study from what FOAMS provides. I fully acknowledge that FOAMS may not have been available when the authors were preparing this manuscript (the FOAMS database was first posted online September 2022, and not published until August 2023), but the FOAMS paper may help emphasize the authors’ points. I am comfortable deferring to the handling editor’s judgment as to whether including FOAMS is appropriate.

Response: Thank you so much for pointing this out; as you correctly assumed, FOAMS was not published at the time this manuscript was written. We are grateful for the FOAMS database and now include in our paper appropriate references to this database and call for future studies to use both datasets. 

Comment 1:

1. Arguments in the manuscript would benefit from reframing:

1.1. Use of self-report:

Introduction paragraph 2 knocks the reliance on self-report measurements – for a variety of valid reasons – and advocates for more objective measures. However, the paradigm the authors employ is entirely self-report, and they discuss this as being a limitation in the Discussion. Thus, this argument in the introduction should be omitted/restructured.

While asking participants to rate the sound of chewing is substantially better than asking them in text how they feel about chewing, it still suffers from response biases, etc. For example, the participants are asked to rate “arousal” on a scale from “calm” to “excited”, and I am interested in how much instruction/coaching was given to understand that question. (Anecdotally, I’ve struggled in my own work with getting participants to think of arousal non-sexually, and I personally don’t know if I’d rate even my most triggering sounds as “exciting”). Further, while the Similarity and Avoidance questions are interesting data points, I’m not convinced they accurately capture whether or not a misophonic person is triggered by that stimulus. (For instance, if I’m triggered by chewing sounds but the particular IADS-2 chewing stimulus doesn’t land for me, how would I answer “To what extent does this sound resemble the sounds in your everyday environment that most bother you?”)

Perhaps, instead, the authors could comment on the wide array of self-reported prompts being asked in misophonia literature (e.g., discomfort, unpleasantness, tolerability, distress, annoyance, etc.) and highlight the utility of asking more standardized questions like valence/arousal, as doing so might help contextualize misophonia in relation to other auditory/psychiatric disorders that have data on valence/arousal.

Response 1: Thank you for these observations and suggestions. We agree that our study is also self-report, and so have removed that section from the introduction. We have also in

---

## [Decision Letter · Decision Letter 1]

4 Mar 2024

PONE-D-23-17567R1Using a Standardized Sound Set to Help Characterize Misophonia: The International Affective Digitized SoundsPLOS ONE

Dear Dr. Trumbull,

Thank you for submitting your manuscript to PLOS ONE. After careful consideration, we feel that it has merit but does not fully meet PLOS ONE’s publication criteria as it currently stands. Therefore, we invite you to submit a revised version of the manuscript that addresses the points raised during the review process. Reviewer 2 is comfortable with the manuscript in its current form but Reviewer 1 made several helpful suggestions regarding the structure of the tables in the manuscript. If the tables are revised according to the Reviewer's suggestions, the manuscript will very likely be accepted.

We look forward to receiving your revised manuscript.

Kind regards,

Andrew R Dykstra

Academic Editor

PLOS ONE

Journal Requirements:

Reviewers' comments:

Reviewer's Responses to Questions

**Comments to the Author**

1. If the authors have adequately addressed your comments raised in a previous round of review and you feel that this manuscript is now acceptable for publication, you may indicate that here to bypass the “Comments to the Author” section, enter your conflict of interest statement in the “Confidential to Editor” section, and submit your "Accept" recommendation.

Reviewer #1: (No Response)

Reviewer #2: All comments have been addressed

2. Is the manuscript technically sound, and do the data support the conclusions?

Reviewer #1: Yes

Reviewer #2: Yes

3. Has the statistical analysis been performed appropriately and rigorously? 

Reviewer #1: Yes

Reviewer #2: Yes

4. Have the authors made all data underlying the findings in their manuscript fully available?

Reviewer #1: No

Reviewer #2: Yes

5. Is the manuscript presented in an intelligible fashion and written in standard English?

Reviewer #1: Yes

Reviewer #2: Yes

6. Review Comments to the Author

Reviewer #1: The authors did a great job responding to reviewer comments. The manuscript is much clearer now, especially throughout the results section. However, in my view the tables are still unnecessarily confusing. Table 2 still has the column heading "valence" even though what's shown in that column are z-scores representing group difference in valence ratings. Perhaps a better way to label that column would be something like "group difference (z-scores)" or something like that? The description in the manuscript text (p. 8) states "higher rankings (e.g., 1, 2) reflect the sounds that most differentiate individuals with misophonia from controls" but this is confusing because there is no actual ranking provided in the table (like 1 or 2) and the numbers in the "valence" column range from ~-4 to 1. Now that the authors removed the language about the IADS-M as the top 10 differentiating sounds, it's also unclear which sounds "count" as part of the IADS-M. Is it all of the sounds in Table 2? Is there a threshold you would recommend for other researchers that distinguishes IADS-M from the rest of the IADS sounds? Tables 3 and 4 are not terribly helpful with the stimuli listed by number and without any description, plus they include information redundant with Table 2. In my view it would be far better to provide an additional two columns for Table 2-- one for each group's mean valence ratings for each stimulus. I don't see why the tables need to include the other dependent measures given that valence was the main variable that differentiated the two groups. If these tables could be made just a little clearer, I do think this paper would be improved and would make a very nice contribution to the literature.

Reviewer #2: The authors did a wonderful job thoroughly considering and responding to each of my comments, and I am happy to recommend this article for publication.

As the authors are finalizing their manuscript, I see one minor piece for the authors to update, which is making sure the citations added in the revision (specifically Orloff et al., 2023; Frank et al., 2020; Simner et al., 2021) are incorporated into the reference list.

Well done!

7. PLOS authors have the option to publish the peer review history of their article (what does this mean?). If published, this will include your full peer review and any attached files.

Reviewer #1: No

Reviewer #2: **Yes: **Heather A Hansen

---

## [Author Response · Author response to Decision Letter 1]

7 Mar 2024

RESPONSE TO REVIEWERS

Response: We appreciate the editors’ and reviewers’ time and consideration of our manuscript. We believe our responses below sufficiently address the concerns and feedback shared. Overall, we believe the changes made have significantly strengthened the manuscript and appreciate your consideration of the revised manuscript for publication in PlosOne.

Reviewer #1: The authors did a great job responding to reviewer comments. The manuscript is much clearer now, especially throughout the results section. However, in my view the tables are still unnecessarily confusing. Table 2 still has the column heading "valence" even though what's shown in that column are z-scores representing group difference in valence ratings. Perhaps a better way to label that column would be something like "group difference (z-scores)" or something like that? The description in the manuscript text (p. 8) states "higher rankings (e.g., 1, 2) reflect the sounds that most differentiate individuals with misophonia from controls" but this is confusing because there is no actual ranking provided in the table (like 1 or 2) and the numbers in the "valence" column range from ~-4 to 1. Now that the authors removed the language about the IADS-M as the top 10 differentiating sounds, it's also unclear which sounds "count" as part of the IADS-M. Is it all of the sounds in Table 2? Is there a threshold you would recommend for other researchers that distinguishes IADS-M from the rest of the IADS sounds? Tables 3 and 4 are not terribly helpful with the stimuli listed by number and without any description, plus they include information redundant with Table 2. In my view it would be far better to provide an additional two columns for Table 2-- one for each group's mean valence ratings for each stimulus. I don't see why the tables need to include the other dependent measures given that valence was the main variable that differentiated the two groups. If these tables could be made just a little clearer, I do think this paper would be improved and would make a very nice contribution to the literature.

Response 1: Thank you for pointing out that we were not clear enough in labeling and explaining the tables! Valence scores for both groups are now in Table 2, and the column heading has been corrected. Tables 3 and 4 have been removed. The IADS-M simply refers to the reordered list, with no particular threshold recommendation- this would be up to researchers to decide how many sounds they would like to use for their purposes. 

Editors Comment 1: As the authors are finalizing their manuscript, I see one minor piece for the authors to update, which is making sure the citations added in the revision (specifically Orloff et al., 2023; Frank et al., 2020; Simner et al., 2021) are incorporated into the reference list.

Response 1: Thank you for pointing out this mistake! The reference list has been amended.

---

## [Editor Report · Decision Letter 2]

12 Mar 2024

Using a Standardized Sound Set to Help Characterize Misophonia: The International Affective Digitized Sounds

PONE-D-23-17567R2

Dear Dr. Trumbull,

We’re pleased to inform you that your manuscript has been judged scientifically suitable for publication and will be formally accepted for publication once it meets all outstanding technical requirements.

Kind regards,

Andrew R Dykstra

Academic Editor

PLOS ONE
---

## [Editor Report · Acceptance letter]

29 Apr 2024

PONE-D-23-17567R2 

PLOS ONE

Dear Dr. Trumbull, 

I'm pleased to inform you that your manuscript has been deemed suitable for publication in PLOS ONE. Congratulations! Your manuscript is now being handed over to our production team.

Kind regards, 

on behalf of

Dr. Andrew R Dykstra 

Academic Editor

PLOS ONE